# Molecular portraits of colorectal cancer morphological regions

Eva Budinská[1], Martina Hrivňáková[1], Tina Catela Ivkovic[2], Marie Madrzyk[2], Rudolf Nenutil[3], Beatrix Bencsiková[3], Dagmar Al Tukmachi[2], Michaela Ručková[2], Lenka Zdražilová Dubská[4], Ondřej Slabý[5], Josef Feit[6], Mihnea-Paul Dragomir[7,8,9], Petra Borilova Linhartova[1], Sabine Tejpar[10], Vlad Popovici[1]*

[1]RECETOX, Faculty of Science, Masarykova Univerzita, Brno, Czech Republic; [2]Central European Institute of Technology, Masarykova Univerzita, Brno, Czech Republic; [3]Masaryk Memorial Cancer Institute, Brno, Czech Republic; [4]Faculty of Medicine, Masarykova Univerzita, Brno, Czech Republic; [5]Central European Institute of Technology, Department of Biology, Faculty of Medicine, Masarykova Univerzita, Brno, Czech Republic; [6]Department of Pharmacology and Toxicology, Faculty of Pharmacy, Masarykova Univerzita, Brno, Czech Republic; [7]Institute of Pathology, Charité-Universitätsmedizin Berlin, Corporate Member of Freie Universität Berlin, Humboldt-Universität zu Berlin, Berlin Institute of Health, Berlin, Germany; [8]Berlin Institute of Health, Berlin, Germany; [9]German Cancer Research Center (DKFZ), German Cancer Consortium (DKTK), Heidelberg, Germany; [10]Faculty of Medicine, Digestive Oncology Unit, Katholieke Universiteit Leuven, Leuven, Belgium

**\*For correspondence:**
vlad.popovici@recetox.muni.cz

**Competing interest:** The authors declare that no competing interests exist.

**Abstract** Heterogeneity of colorectal carcinoma (CRC) represents a major hurdle towards personalized medicine. Efforts based on whole tumor profiling demonstrated that the CRC molecular subtypes were associated with specific tumor morphological patterns representing tumor subregions. We hypothesize that whole-tumor molecular descriptors depend on the morphological heterogeneity with significant impact on current molecular predictors. We investigated intra-tumor heterogeneity by morphology-guided transcriptomics to better understand the links between gene expression and tumor morphology represented by six morphological patterns (morphotypes): complex tubular, desmoplastic, mucinous, papillary, serrated, and solid/trabecular. Whole-transcriptome profiling by microarrays of 202 tumor regions (morphotypes, tumor-adjacent normal tissue, supportive stroma, and matched whole tumors) from 111 stage II-IV CRCs identified morphotype-specific gene expression profiles and molecular programs and differences in their cellular buildup. The proportion of cell types (fibroblasts, epithelial and immune cells) and differentiation of epithelial cells were the main drivers of the observed disparities with activation of EMT and TNF-α signaling in contrast to MYC and E2F targets signaling, defining major gradients of changes at molecular level. Several gene expression-based (including single-cell) classifiers, prognostic and predictive signatures were examined to study their behavior across morphotypes. Most exhibited important morphotype-dependent variability within same tumor sections, with regional predictions often contradicting the whole-tumor classification. The results show that morphotype-based tumor sampling allows the detection of molecular features that would otherwise be distilled in whole tumor profile, while maintaining histopathology context for their interpretation. This represents a practical approach at improving the reproducibility of expression profiling and, by consequence, of gene-based classifiers.

## eLife assessment

This study presents a **valuable** finding on the putative molecular patterns underlying characteristic morphological regions observed in colorectal cancer (CRC). The authors provide a morphological framework through which clinicians might improve the performance of molecular signatures and consequently predict the clinical response of patients with better accuracy. The evidence supporting the claims of the authors is **solid**. The work will be of interest to clinicians and cancer biologists working in the field of CRC.

## Introduction

Colorectal cancer (CRC), the third cause of death among cancer patients, is a highly heterogeneous disease, with a slow initial progression that favors the accumulation of mutations leading to a complex phenotype. Differences that exist both between and within tumors of the same cancer type are a major hurdle towards proper treatment selection and for developing more targeted therapies. Depending on the perspective under which these differences are investigated, various categorization paradigms have emerged. The systematization of clinical and histopathological parameters led to the definition of current TNM staging system (*Amin, 2017*), which presently constitutes the gold standard for diagnosis and prognosis. The development of high throughput molecular technologies brought a novel perspective and set the stage for the appearance of molecular taxonomies categorizing the tumors into subgroups sharing common molecular traits (*Perez-Villamil et al., 2012*; *Budinska et al., 2013*; *Marisa et al., 2013*; *De Sousa E Melo et al., 2013*; *Sadanandam et al., 2013*; *Roepman et al., 2014*) with consensus molecular subtypes (CMS)(*Guinney et al., 2015*) representing their common denominator. While these studies were based on whole-tumor (bulk) gene expression data, the developments in single-cell sequencing further refined the CMS classes adding two intrinsic epithelial subtypes (iCMS2/3) to the picture (*Joanito et al., 2022*). Other studies combined genomics and transcriptomics data and an alternative classification emerged (*Muzny et al., 2012*).

Whole transcriptome expression profiling of tissue sections is generally performed on RNA extracted from regions of interest covering diverse cell collections. By consequence, the expression levels associated with various transcripts represent, in the end, a weighted mean of contributions of each cell type, being driven by the most abundant ones. The signals from less abundant cell types are reduced or even silenced and are, therefore, overlooked. In the case of solid tumors, this approach requires a representative region, enriched in tumoral cells, to be selected in the tissue section(s) and used for RNA extraction. This is the predominant approach to tissue expression profiling that fueled the myriad of studies over the last two decades and led to significant progress in understanding the various cancers. Newer technologies such as single cell sequencing and spatial transcriptomics allow for a much finer selection of cells to be interrogated (*Tang et al., 2019*; *Rao et al., 2021*). However, while powerful, these techniques rely on fresh tissue and have still to find their place in routine clinical practice.

The importance of a morphological perspective on the molecular classification has been acknowledged from the beginning, *Jass, 2007* already identifying several morphological features associated with the five groups proposed (e.g. serration, mucinous and poor differentiation were highly present in two of the five groups), but also noted that these features were not sufficient for predicting the groups. Later, *Budinska et al., 2013* proposed six morphological patterns (morphotypes) as major histological descriptors and showed that a two-tier histological score is strongly associated with the five molecular subtypes identified. Interestingly, a pure data-driven image-based classifier for the same molecular subtypes resulted in selecting remarkably similar morphological motifs (*Budinska et al., 2016*; *Popovici et al., 2017*). *Müller et al., 2016* reviewed the TCGA and CMS subtypes and their links with some morphological aspects, most notably the serrated phenotype. It is worth mentioning that in all these cases the evaluation of the morphological features referred to the whole tumor section; for example, a tumor was considered of mucinous morphology if the mucinous pattern was present in more the 50% of the tumor region, in accordance with standard definitions endorsed by the World Health Organization (*Bosman, 2010*). These links between tumor morphology and molecular features also imply that the gene expression profile may depend on the tumor region sampled for RNA extraction. The sensitivity of gene-based classifiers to tumor sampling raised concerns regarding

the stability of consensus molecular subtypes (*Dunne et al., 2016*) and may partially explain the low proportion of biomarkers that reach clinical relevance (*Stewart et al., 2017*).

It is evident that, while intra-tumoral heterogeneity is recognized as a major challenge, we still lack the practical tools for its characterization that would easily translate into a diagnostic and predictive model. In contrast with previous results, in our study we explored region- (morphotype-) based transcriptomics approach as a possible solution to this problem. This method offers a trade-off between whole-tumor profiling and spatial transcriptomics. It has a better signal resolution than whole-tumor profiling, since it selects tumor regions with more similar cellular buildup, and covers the whole transcriptome, but clearly has a much lower spatial resolution than true spatial transcriptomics. However, it represents a practical approach where several regions of interest can be stably identified, and their profiling could be easily integrated in the current molecular pathology diagnostic practice.

Building on our previous results (*Budinska et al., 2013*), we based our study on a detailed exploration of the transcriptome of the six morphotypes identified earlier as associated to the molecular subtypes of CRC: complex tubular (CT), desmoplastic (DE), mucinous (MU), papillary (PP), serrated (SE) and solid/trabecular (TB), respectively. As reference, we also profiled several tumor-adjacent normal (NR) and supportive stroma (ST) regions. The present study was based on a single center cohort and was designed to achieve several goals: (i) identifying representative samples for each of the morphotypes, (ii) providing a comprehensive characterization of their transcriptomics landscape, and (iii) studying the intra-tumoral heterogeneity from the perspective of morphotype-resolved transcriptomics. We characterized the morphotypes from several transcriptional angles: basic molecular programs as captured by differential expression and pathway analyses, molecular tumor classifiers and prognostic gene signatures. At the same time, we looked for variations both across all tumors and across matched samples (within tumors). The emerging picture is of an unexpectedly high heterogeneity, with clear implications both from fundamental biological and practical perspectives, opening new avenues for biomarker design.

## Results
### Data

From n=111 unique cases of primary CRC tumors (stages: II: 59, III:32, IV:20), n=202 regions were macrodissected representing either tumor morphological regions (n=149), tumor-adjacent normal tissue (NR, n=17), supportive stroma (ST, n=8), or whole tumor (n=28), respectively. Among the tumor morphological regions, n=126 'core samples' were identified based on 'morphological purity', indicating regions containing at least 80% of a unique morphological pattern. The six morphotypes of interest (*Figure 1*) consisted of (in brackets the additional non-core samples) 41 (+11) CT, 13 (+2) DE, 18 (+3) MU, 10 (+2) PP, 33 (+7) SE, and 9 TB samples, respectively. The distribution of associated main clinical parameters is given in *Supplementary file 1*. The only statistically significant associations found were between MU or TB and grade 3 tumors, and SE and lower grade tumors (p=0.019, *Supplementary file 2*), respectively.

To complement the results presented here, we created a web application https://morphogene. recetox.cz allowing the interrogation of gene expression in various morphological regions.

### Morphotype cellular admixtures

The transcriptomic profile of solid tumor sample is a mixture of gene expression profiles of individual cell types and their specific programs, including cancer cells at different levels of differentiation, specific immune cells, or supportive fibroblasts. As a first step, we performed in-silico deconvolution of the expression profiles to identify the most prevalent cell types in each of the morphotypes and GSEA to score cell-type-specific gene sets (see Materials and methods) in each morphotype, and NR and ST regions (used as controls, *Figure 2*, *Supplementary files 3–4*).

The results from ESTIMATE indicated, as expected, a high stromal content for ST, DE, and MU and a high epithelial tumor cell content for normal region and TB and SE morphotypes, respectively (*Figure 2A*). A more balanced situation was observed for CT and PP morphotypes (similar to NR). This agreed with the (stroma-related) 'Isella signatures' (*Isella et al., 2015*) where ST, DE, and MU were enriched in endothelial cells, CAFs and immune cells (*Figure 2B*). When investigating the categories of epithelial cells, the signatures of top of the normal colon crypt cells (*Kosinski et al., 2007*) and

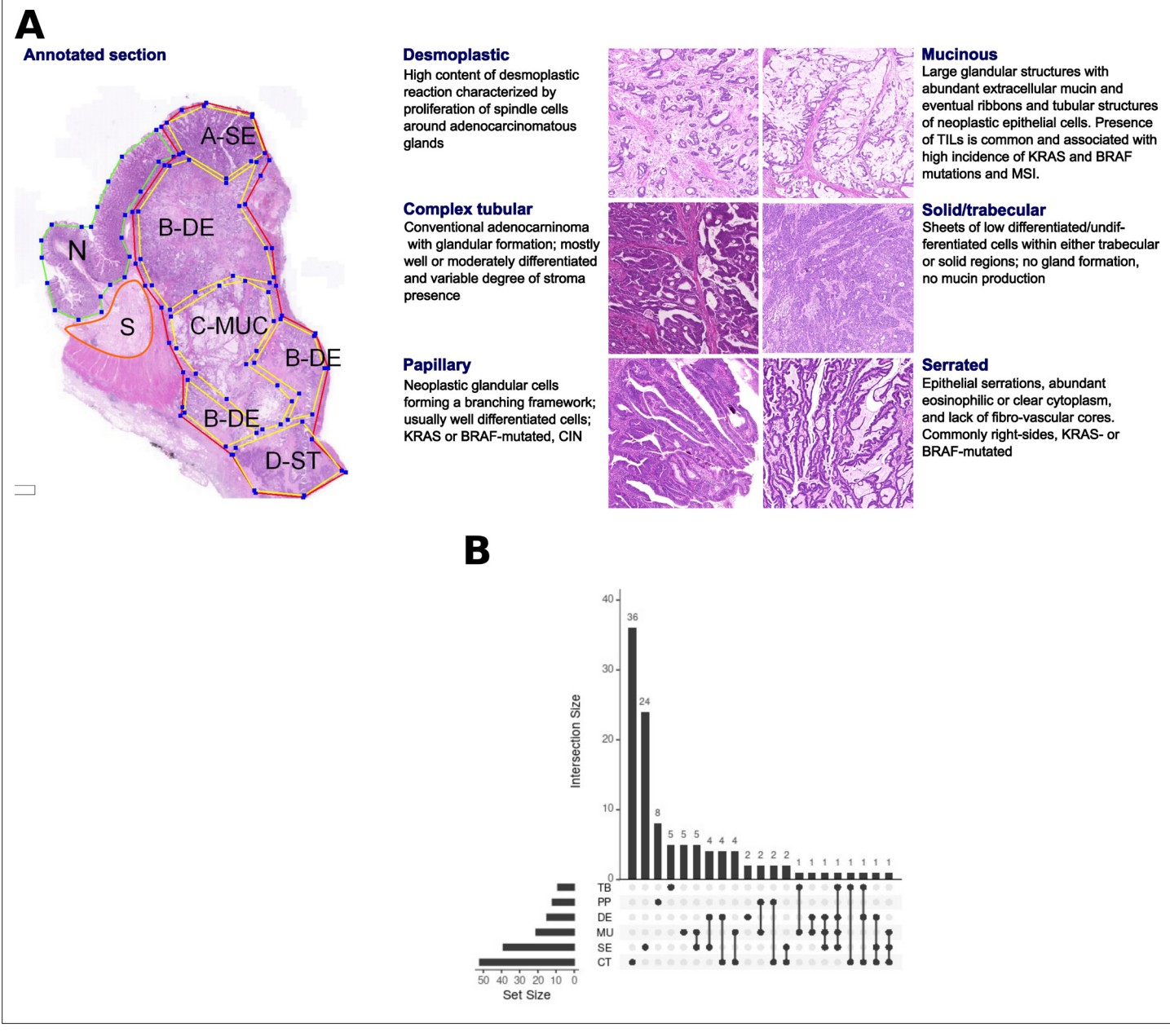

**Figure 1.** Morphological patterns and their distribution in the dataset. (**A**) The six CRC morphological patterns of interest (morphotypes). *Left*: example of an original annotation used for macrodissection and RNA extraction. Note that the original annotations in the image are not identical to the ones used in the main text. Here, A-SE stands for serrated (SE) in the text, B-DE for desmoplastic (DE) in the text, C-MUC for mucinous (MU) in the text, and D-ST for solid/trabecular (TB) in the text, respectively. Also, N indicates a tumor-adjacent normal epithelial region and S a supportive stroma region, respectively. *Right*: examples of morphotypes – complex tubular (CT), desmoplastic (DE), mucinous (MU), papillary (PP), serrated (SE), and solid/trabecular (TB). (**B**) Morphotype distribution per case (unique tumor) and intersections thereof: some cases had several morphotypes profiled.

colon differentiated epithelial cells (*Merlos-Suárez et al., 2011*) were enriched solely in NR regions, while DE, MU, CT, SE, and TB were depleted in these cell types (*Figure 2B*). On the other hand, MU, CT, PP, and TB regions expressed genes specific for the basal crypt cells (*Kosinski et al., 2007*) and ST, DE, and MU were enriched in signatures of intestinal stem cells. These observations are in perfect agreement with the definition of the morphotypes and confirm the proper selection of the samples. quanTIseq revealed that all tumor morphotypes were enriched in M1 macrophages (with maximal presence in MU and DE), while M2 macrophages, NK cells and myeloid dendritic cells where

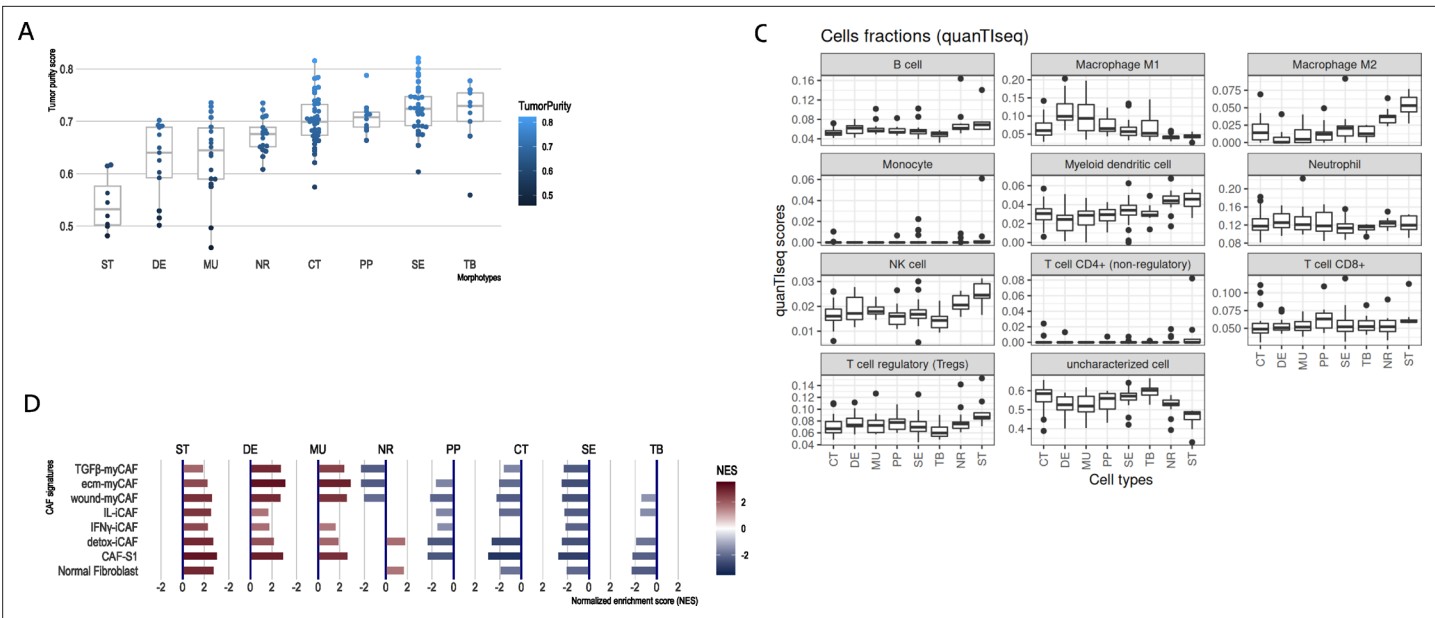

**Figure 2.** CRC morphotypes: in silico decomposition of the cellular admixture. (**A**) Boxplots of the tumor purity (epithelial content – ESTIMATE method) in each tumor morphotype and the two non-tumor regions, ordered by increasing median values. (**B**) Signatures specific to colon crypt compartments and major cell types estimated from gene expression data in terms of normalized enrichment scores (NES): only statistically significant scores are shown. (**C**) Immune cell fractions (and unassigned fractions) inferred from gene expression data using quanTIseq method. (**D**) Types of cancer-associated fibroblasts (CAFs) as estimated from gene expression using the signatures from *Khaliq et al., 2022*; *Kieffer et al., 2020*.

The online version of this article includes the following figure supplement(s) for figure 2:

**Figure supplement 1.** Epithelial signatures from *Pelka et al., 2021*.

**Figure supplement 2.** Immune signatures from *Pelka et al., 2021*.

**Figure supplement 3.** Stromal signatures from *Pelka et al., 2021*.

highly present in supporting stroma and tumor-adjacent normal regions (*Figure 2C*). Additionally, TB morphotype had the lowest scores for regulatory T cells (TREGs) and B cells.

Further we refined the morphotype cell admixtures by testing signatures of different cell types and their active programs as derived from single-cell sequencing studies. We evaluated more than 150 signatures of stromal, epithelial, and immune cell population (supplemental tables of *Pelka et al., 2021*) and cancer associated fibroblasts (CAFs) (*Khaliq et al., 2022*; *Kieffer et al., 2020*) (see *Supplementary files 3–4* for full signatures). Interestingly, the morphotypes differed in the signatures of CAFs subpopulations (*Figure 2D*). ST, MU, and DE had high GSEA scores of most of the CAFs subpopulations, while the rest (CT, PP, SE, and TB) had mostly negative scores, indicative of depletion of corresponding cell types. DE and MU were most strongly enriched in signatures of ECM-myCAF S1 – associated with immunosuppressive microenvironment and pro-metastatic functions (*Kieffer et al., 2020*) – and wound-healing myCAF S1 populations, while the adjacent stroma mainly showed signatures of normal fibroblasts, detox-iCAF S1 and IL-iCAF S1 populations, both characterized by detoxification and inflammatory signaling. NR regions were enriched only in normal fibroblasts and detox-iCAF S1. By exploring the signatures from *Pelka et al., 2021*, we observed even finer differences between morphotypes within all three cell type populations and their programs (*Figure 2— figure supplements 1–3*). For instance, CT, TB, and SE had enriched pS04 (ribosomal) and pS12 (proliferation) stromal cell signatures, in addition, CT and TB expressed pS05 (interferon-stimulated genes, ISGs) and pS21 (*FOS*, *JUN*) signatures. Also, NR had a specific enrichment in mitochondrial (pS09), metallothionein (pS16) and BMP-producing (pS17) fibroblasts. CT and TB resembled MU in expressing pS20 signature and, additionally, TB showed similar levels of pS13 (inflammatory) signature as MU and DE. ST regions and DE and MU morphotypes had significantly increased pS02 (Fibro. matrix/stem cell niche) signature. Full results for other cell types and programs are provided in the *Supplementary file 4*.

**Table 1.** Results of comparison of each morphotype (and the two non-tumoral regions) with the average profile.

The table shows the top 20 up- and down- regulated genes and significantly activated hallmark pathways and processes (as result of GSEA). The genes not significant after p-value adjustment (at FDR = 0.15) have their symbols greyed. See also *Supplementary files 5–6*.

| Morph | Top 20 up-regulated genes (compared to mean) | Top 20 down-regulated genes (compared to mean) | Hallmark pathways with high score | Active processes (based on the active hallmark pathways) |
|---|---|---|---|---|
| MU | ARF4, MUC2, SULF1, FNDC1, LOXL1, LGALS1, ANTXR1, BGN, COL12A1, PALLD, MEG8, DKK3, ACVR1, GPX8, CALD1, FBN1, MLLT11, CSRP2, TUSC3, GREM1 | TIMD4, PRELID3BP3, EREG, KDM4A, CCDC175, TDP2, CHMP1B2P, ACE2, NLRP7, UGT2A3, SLC26A3, A1CF, TSPAN6, CLDN10, TMIGD1, BMP5, MS4A12, FAM3B, CLCA4, MEP1A | EMT, TNF a signaling via NFKB, Complement, IL2 STAT5 signaling, hypoxia, inflammatory response, KRAS signaling, UV response, myogenesis, coagulation, apical junction, allograft rejection, IL6 JAK STAT3 signaling, interferon gamma response, apoptosis, TGF-beta signaling, angiogenesis, hedgehog signaling, estrogen response early, NOTCH signaling, WNT beta catenin signaling, cholesterol homeostasis | Inflammation, neoangiogenesis, increased metastatic potential, apoptosis, development |
| DE | OLFML2B, INHBA, LUM, SULF1, PTPN14, PRDM6, SPOCK1, RDX, EDNRA, COL12A1, CTHRC1, PRRX1, LGALS1, COPZ2, COL10A1, TNFAIP6, IGFL1P1, ST6GAL2, FAP, BGN | SLC17A4, ANPEP, DEFA5, RAP1GAP, MRAP2, ADH1C, TRIQK, REG1A, SLC4A4, UGT2B15, REG4, SEMA6A, L1TD1, MS4A12, SI, SPINK4, CLCA4, MUC2, CLCA1, CA1 | EMT, TNF a signaling via NFKB, Complement, IL2 STAT5 signaling, hypoxia, inflammatory response, KRAS signaling, UV response, myogenesis, coagulation, apical junction, apoptosis, TGF-beta signaling, angiogenesis, hedgehog signaling, estrogen response early | Inflammation, neoangiogenesis, increased metastatic potential, apoptosis |
| PP | PTPRD, KNDC1, MIMT1, UPK3B, MPZ, MMP15, CYP4F12, SNORD4A, SNAR-C3, TMTC4, LRCOL1, GATA5, SNAR-E, EPHA7, IPO4, SNAR-I, CASC21, NUTF2, SNAR-B2, RPL31P50 | IGKV3-11, IGHV4-39, ANPEP, OR4F8P, HEPACAM2, ADAM28, CPS1, TMIGD1, NPY6R, ITLN1, SI, ADH1C, CAV1, MMP2, FDCSP, CLU, REG1A, RSPO3, PAX8-AS1, PALMD | MYC targets V1, MYC targets V2, E2F targets, KRAS signaling DOWN, WNT beta catenin signaling, | |
| SE | PPAN-P2RY11, TUBB4BP7, JADE3, PFDN6, CLDN2, YAF2, BOLL, SLAMF9, SLC12A2, CCDC175, GRIN2B, TUBB3P2, GAPDHP71, RPS2P25, MAT1A, NOX1, SNORD12C, SMAD6, MECOM, EXTL2 | IGKV2D-29, MYLK, TAGLN, CNTNAP3P2, GLI3, CPXM2, NR3C1, CNN1, PECAM1, COLEC12, IGKV4-1, IGKV2D-30, DPYD, CLU, TSHZ2, ADH1B, IL10RA, PDE7B, ABCA8, CDC42SE2 | MYC targets V1, MYC targets V2, E2F targets, G2M checkpoint, | |
| CT | TMEM97, RPL13, CLDN1, TFDP1, CKS2, CDCA7, TPX2, ANLN, RAD54B, KRT18, HSPH1, CCT6A, PLK1, TMEM97P2, CSE1L, MIPEP, SNORA71D, SNORA71C, PTTG1, PLBD1 | CR2, OGN, SNORD114-21, SLC30A10, CLCA4, SNORD114-12, DCLK1, FAT4, CPA3, ADH1B, SLC26A2, SNORD114-20, SFRP1, ZG16, FGF7, SNORD113-1, ABCA8, B4GALNT2, MS4A12, CA1 | MYC targets V1, MYC targets V2, E2F targets, G2M checkpoint, MTORC1 signaling, unfolded protein response, Glycolysis, oxidative phosphorylation, fatty acid metabolism, protein secretion | Proliferation, Catabolism, oxidative stress, cell cycle disruption |
| TB | CKAP2, HSP90AA1, PPP3CA, REEP4, MSH6, TOP2A, HSPE1, PPP2R5C, TBCA, VRK2, NIFK, TXNL4A, MNAT1, ERI1, XPO1, VTRNA1-2, ANP32A, ARF6, RNF2, EIF4A1P7 | FLJ22763, TMEM236, NPY6R, IGKV3D-20, IGKV2D-30, OLFM4, SELENBP1, LRRC19, CDHR1, IGHA1, SNORD123, SLC26A3, CXCL14, SLC3A1, SEMA5A, MS4A12, IGHA2, CLCA4, NXPE4, NXPE1 | MYC targets V1, MYC targets V2, E2F targets, G2M checkpoint, MTORC1 signaling, unfolded protein response, Glycolysis, oxidative phosphorylation, fatty acid metabolism, protein secretion, cholesterol homeostasis, | Inflammation, catabolism, apoptosis, oxidative stress, proliferation, cell cycle disruption |
| NR | PIGR, SLC26A3, ADH1B, NXPE1, IGHA2, CLCA1, JCHAIN, IGHA1, FCGBP, IGK, NXPE4, SLC9A2, MUC2, NR3C2, TMEM236, MS4A12, FABP1, IGLC3, IGKV1D-39, LRRC19 | TACSTD2, FAM83D, ASPN, CXCL11, CTHRC1, SLC39A6, IFNE, SULF1, HSPH1, ELFN1-AS1, THBS2, CLDN1, SIM2, SLC22A3, SPARC, FN1, AHNAK2, COL11A1, SPP1, INHBA | Heme metabolism, bile acid metabolism, xenobiotic metabolism, fatty acid metabolism | |
| ST | SFRP2, ADH1B, EMCN, STEAP4, ADAMTS1, ABI3BP, SPARCL1, DCN, PTGDS, PALMD, NOVA1, SLIT3, OGN, SERPINF1, RSPO3, CPA3, FBLN5, C3, EFEMP1, PBX3 | FRK, AADACP1, CKS2, HOOK1, CLDN1, ANLN, S100P, UGT8, MACC1, EXPH5, CYP3A5, OCIAD2, SLC12A2, GK, EVADR, TMC5, REG4, TFF1, TCN1, CXCL8 | EMT, TNF a signaling via NFKB, Complement, IL2 STAT5 signaling, hypoxia, inflammatory response, KRAS signaling, UV response, myogenesis, coagulation, apical junction, allograft rejection, IL6 JAK STAT3 signaling, interferon gamma response | Inflammation, neoangiogenesis, increased metastatic potential |

## CRC morphotypes and molecular programs

The molecular programs and pathways represented in MSigDB were scored by performing GSEA on differentially expressed genes (DEGs) in all morphotypes (and NR and ST).

For the first analysis, the ordered lists of DEGs per morphotype were obtained by contrasting the individual expression profiles to the average profile of pooled samples (*Supplementary file 5* contains all DEGs). This allowed the identification of all molecular programs significantly de-/activated in each morphotype (*Table 1*, *Figure 3*; *Supplementary file 6*). When considering only the hallmark signatures (H collection), the discriminative gradients between the morphotypes (and NR and ST) were along the EMT and TNF-α signaling axes at one end, and the *MYC* and E2F targets at the other end (*Table 1*, *Figure 3A*). Desmoplastic and mucinous shared active pathways involved in immune system response (TNF-α signaling via NF-κB, interferon gamma response, complement, *IL2-STAT5* signaling), neoangiogenesis, and increased metastatic potential (EMT, coagulation, TGF-β, NF-kB, NOTCH, Apical junction). At the other end of the spectrum, CT and TB morphotypes had activated major pathways involved in proliferation processes (P53, MTORC 1, Myc targets, G2M checkpoint, Mitotic spindle, NOTCH signaling, Protein secretion). In contrast with CT, TB morphotype shared with MU and DE active TGF-β signaling, apoptosis, and most pathways involved in immune system response. PP and SE morphotypes had activated *MYC* and E2F targets, with PP morphotype exhibiting downregulation of the *KRAS* signaling and upregulation of the WNT-β catenin signaling.

We performed principal component analysis (PCA) of the GSEA scores of hallmark pathways. Their projection onto the first two principal components revealed a specific bi-dimensional clustering of the morphotypes and illustrated the gradient of changes between morphotypes (*Figure 3B*, *Figure 3—figure supplement 1*). At one end, MU and DE shared the same region in PCA space with positive coordinates on the axis defined, among others, by EMT, inflammatory response, and UV response. At the same time, they had opposite projections on the second axis of variation, defined by p53, unfolded protein response and cholesterol homeostasis. In contrast, SE and PP shared the same quadrant with negative coordinates on the first axis, but positive on the second axis. The CT and TB fell between the two previous groups with respect to the first axis of variation, while having similar activations of pathways defining the principal components. Overlaid on top of the transcriptomics layer, an additional gradient could be observed: epithelial cell differentiation. Indeed, while SE, PP, and CT were well or moderately differentiated, TB, DE and MU had low or undifferentiated morphology.

*Figure 3C* shows heatmap of median expressions of all top 5 up- and down-regulated genes of each morphotype with respect to the average profile (full lists in *Supplementary files 5–6*). A second analysis identified morphotype-specific processes and pathways by GSEA of differentially expressed genes between each morphotype and all other five (excluding ST and NR) (*Supplementary files 7–8*).

Several macrodissected regions originated from the same section allowing for paired comparison of morphotypes. While the reduced number of such pairs (MU vs SE: 8 pairs, DE vs SE: 7, CT vs DE: 5, and CT vs MU: 5, respectively) impacted the statistical power, we were able to identify genes differentially expressed (after p-value adjustment) in all but MU vs SE, indicative of regional differences (*Supplementary files 9–10*). The differences between gene expression signatures from the matched paired comparisons were in line with those from comparisons not accounting for sample pairing, indicating that the morphotype specific effect was dominating the contrasts (see *Figure 3—figure supplement 2*).

We also performed comparison between all pairs of morphotypes (*Supplementary files 11–12*). This comparison shows that, despite similar content in terms of fibroblasts or epithelial cells (discussed above), there are still differences both in terms of differentially expressed genes (*Supplementary file 11*) and activated molecular programs (*Supplementary file 12*) between DE and MU, on one side, and CT, PP, SE, and TB. These results refine those presented above and allow an ordering of morphotypes in terms of relative activation of pathways. For example, *KRAS* signaling appears to be highest in PP, followed by CT.

## Morphotypes and molecular subtypes

The molecular subtyping taxonomies of CRC were derived from datasets representing profiles of whole tumor sections, therefore aggregating the expression of many cell types. In our previous work (*Budinská et al., 2013*), we associated molecular subtypes with morphotypes assessed on the whole tumor and hence we were interested to see how this observation translated to the case of

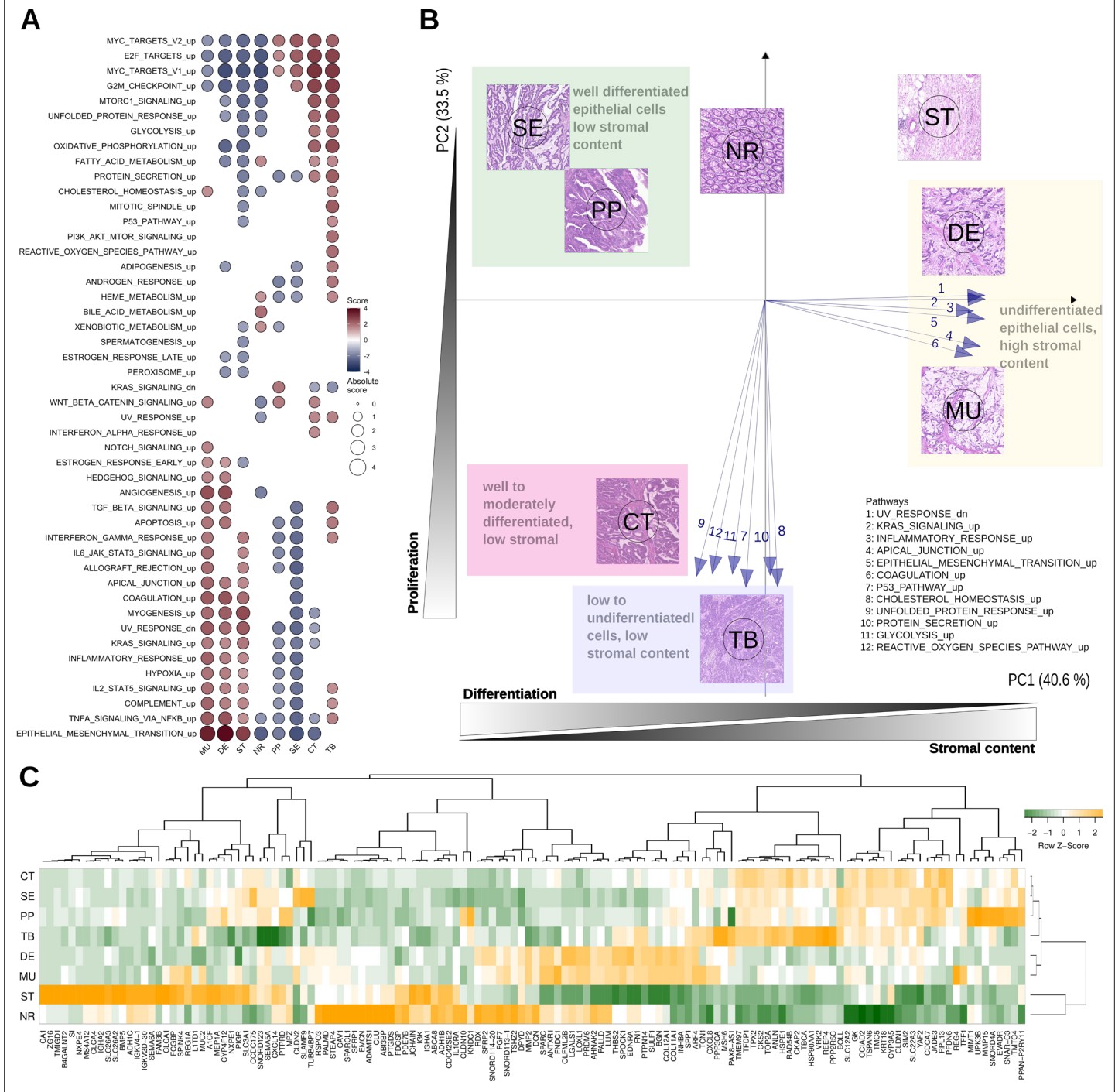

**Figure 3.** Top differentially expressed genes and hallmark pathways. (**A**) GSEA scores for hallmark pathways in the six morphotypes and two non-tumoral regions. Only pathways with statistically significant scores are shown. (**B**) Principal component analysis of hallmark pathways: the median profiles of the six morphotypes (CT: complex tubular, DE: desmoplastic, MU: mucinous, PP: papillary, SE: serrated, and TB: solid/trabecular) and the two non-tumoral regions (NR: tumor-adjacent normal and ST: supportive stroma) are projected onto the space defined by first two principal components (74% of the total variance). The top pathways contributing to the principal axes are shown as well. See also *Figure 3—figure supplement 1*. (**C**) Heatmap of top 5 up- and down-regulated genes for each of the six morphotypes.

The online version of this article includes the following figure supplement(s) for figure 3:

**Figure supplement 1.** Principal component analysis of hallmark pathways GSEA scores: loadings for the first two principal components, i.e., contribution of pathways to the first two axes.

**Figure supplement 2.** Hallmark pathways differential activation between pairs of morphotypes.

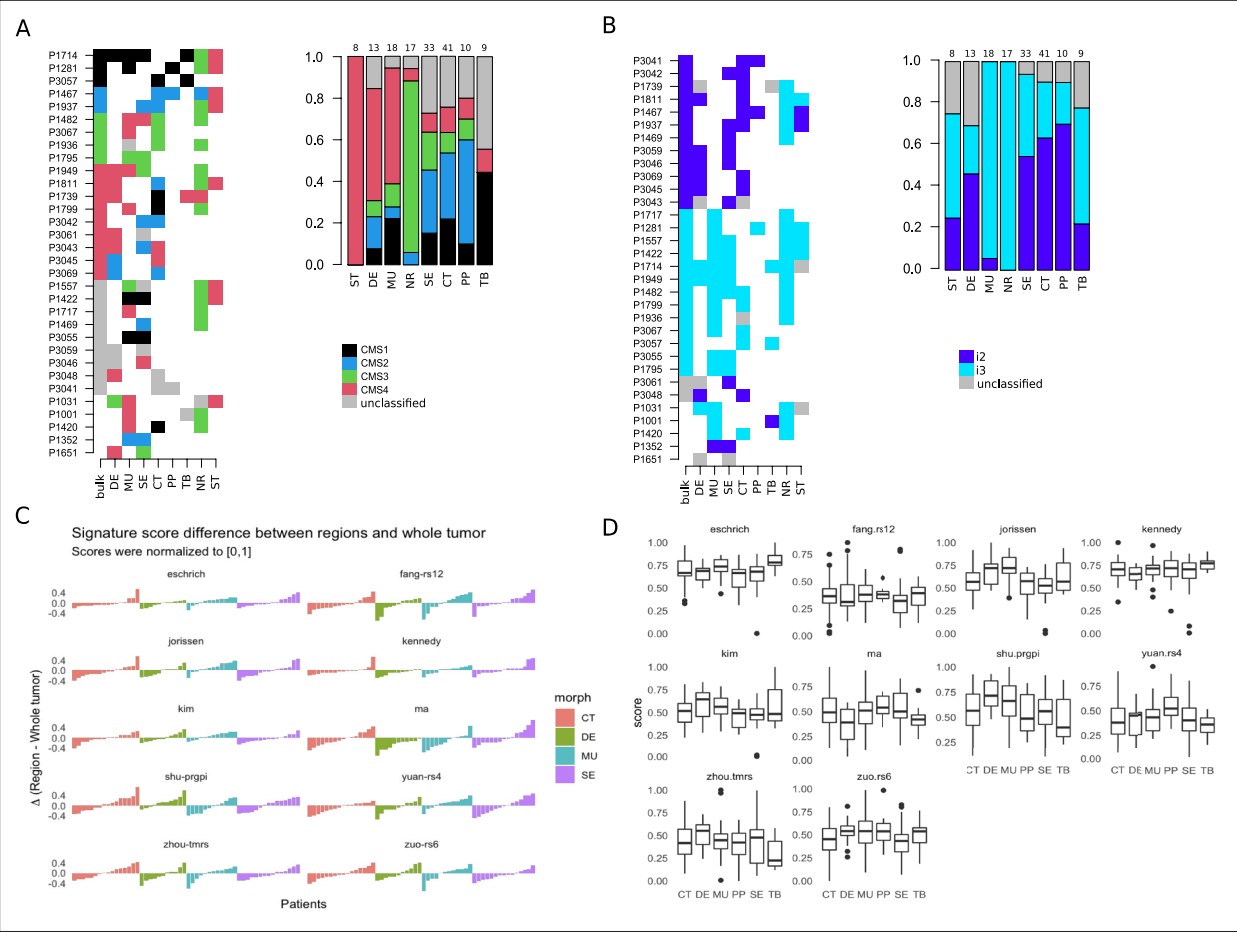

**Figure 4.** Intra-tumoral heterogeneity and the morphotypes (for all core samples, including those unassigned by the classifiers). Only cases with at least two distinct morphotypes present are shown. (**A**) Left: CMS assignment for tumors represented by multiple regions. Right: CMS assignment per morphotype (and two non-tumoral patterns). (**B**) Left: iCMS assignment for tumors represented by multiple regions. Right: iCMS assignment per morphotype (and two non-tumoral patterns). (**C**) Differences between paired signatures: morphotypes vs whole tumor (each signature was normalized to [0,1] prior to computing the differences). Only four (morphotype, whole tumor) pairs were represented enough in the data. (**D**) Boxplots for the ten (normalized) signatures across morphotypes. The 'Eschrich' and 'Jorissen' signatures vary significantly (Kruskal-Wallis's test) across morphotypes. For equivalent plots for all samples, including non-core, see *Figure 4—figure supplement 1*.

The online version of this article includes the following figure supplement(s) for figure 4:

**Figure supplement 1.** Molecular subtypes and morphotypes in all samples, including non-core samples.

macrodissected morphological regions. We predicted both the consensus (CMS) (*Guinney et al., 2015*) and intrinsic (iCMS) (*Joanito et al., 2022*) molecular subtypes.

All ST regions were predicted as CMS4, and 82.4% of NR regions as CMS3. For the morphotypes, the predictions were more distributed across subtypes: DE and MU were most often assigned to CMS4 (63.6% and 58.8%), PP, SE, and CT to CMS2 (62.5%, 41.7% and 41.9%) and TB to CMS1 (80%; *Figure 4A*, *Figure 4—figure supplement 1*). More importantly, this heterogeneity was also observed intra-tumoral, with regions within the same tumor section being assigned to different subtypes (*Figures 4A and 5*, *Figure 5—figure supplements 1 and 2*).

In contrast, intrinsic molecular subtypes (iCMS2/3) were much more stable, most of the time all the morphotypes within a tumor sharing the same iCMS label (*Figure 4B*, *Figure 4—figure supplement 1*) and agreeing with the whole-tumor assignment. NR, MU, TB, and ST regions were classified most of the time as iCMS3 (100%, 94,4%, 71.4%, 66.7%), while PP, CT and DE were predominantly classified as iCMS2 (77.8%, 70.3%, 66.7%). The serrated morphotype was almost equally assigned to each of the iCMSs (iCMS2: 58%, iCMS3:42%).

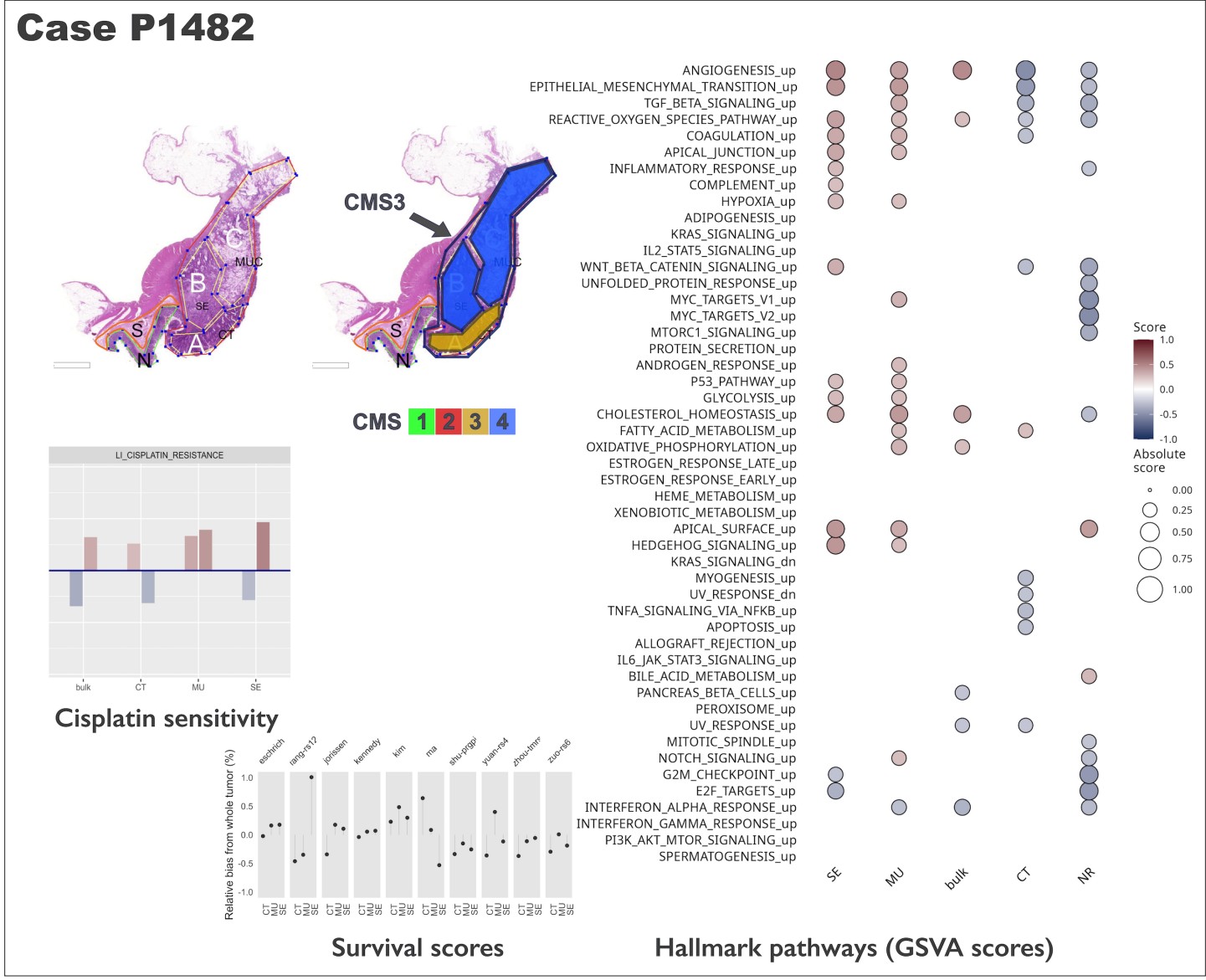

**Figure 5.** Intra-tumoral heterogeneity case study. For the same case, different CMS labels are assigned to regions and whole tumor profile. The hallmark pathways show various levels of activation (as computed by GSVA) within same section. The relative change in prognostic scores indicate potential underestimation of risk for some signatures, while others appear to be stable across tumor. See also *Figure 5—figure supplements 1 and 2*. Note that in the pathology section image, the original annotations were preserved, and they are not identical to the ones used in the main text. Here, MUC stands for mucinous (MU) in the text. Also, N indicates a tumor-adjacent normal epithelial region and S a supportive stroma region, respectively.

The online version of this article includes the following figure supplement(s) for figure 5:

**Figure supplement 1.** Intra-tumoral heterogeneity additional case study.

**Figure supplement 2.** Intra-tumoral heterogeneity additional case study.

## Prognostic and predictive gene-based signatures

The morphotypes generally differed in terms of score distributions, with two signatures reaching statistical significance (Kruskal-Wallis's test: Eschrich p=0.0228, Jorissen p=0.00085, *Figure 4C–D*). A more pronounced variability was observed when comparing tumor regions to matched whole tumor, with amplitude of the differences (region vs whole tumor) larger than 50% of the whole tumor score in some cases (*Figure 4C*). *Figure 5* shows a case study with three different morphological regions (CT, MU, SE) which manifest rather large deviations from the whole tumor-based risk scores for most of the prognostic signatures (see also *Figure 5—figure supplements 1 and 2*).

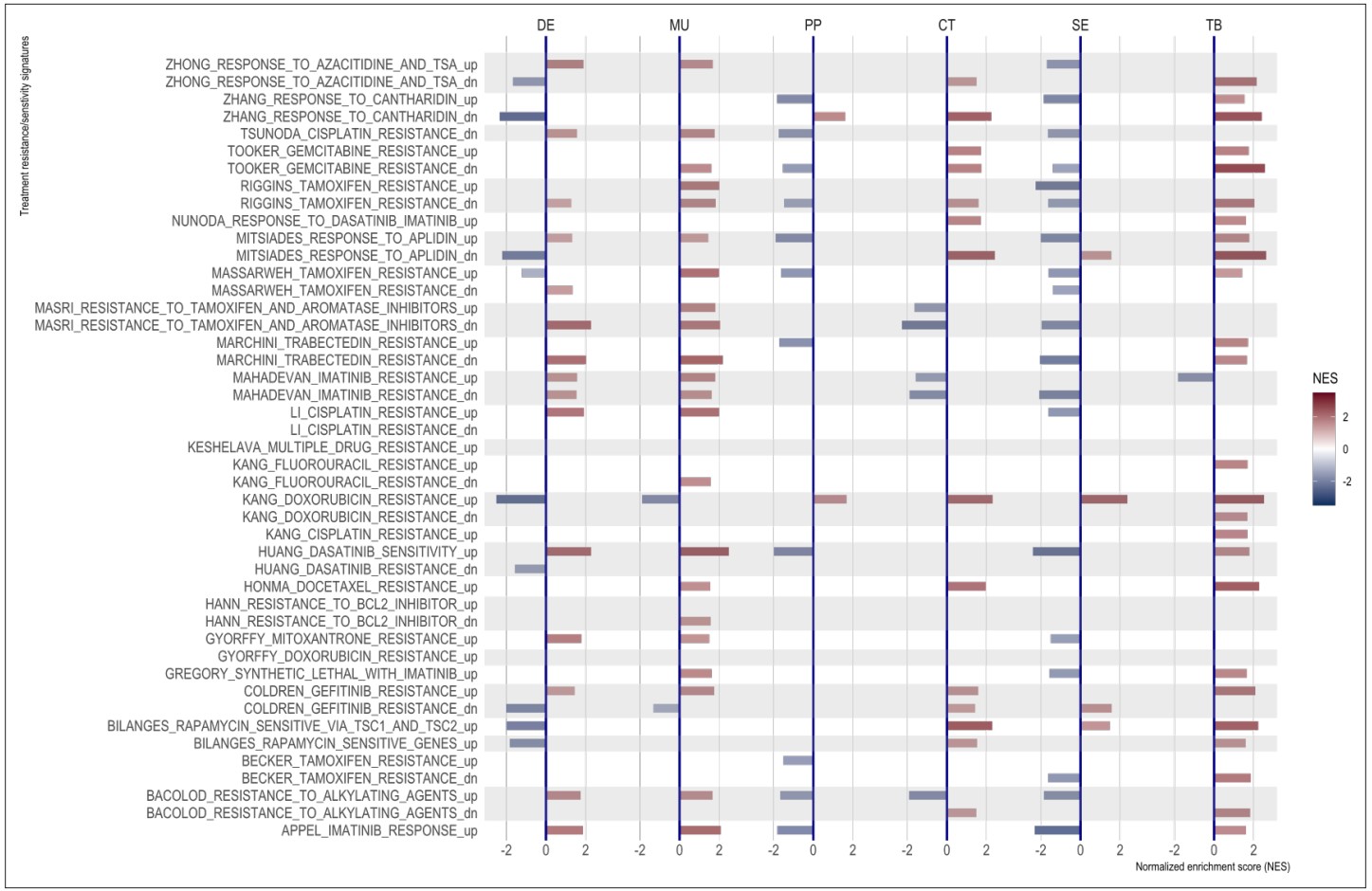

**Figure 6.** Normalized enrichment scores from GSEA for selected resistance signatures (from C2 section of MSigDB). Only significant scores are shown.

The online version of this article includes the following figure supplement(s) for figure 6:

**Figure supplement 1.** Resistance scores (GSVA) per patient and morphotype for cases where the whole–tumor prediction is contradicted by some regional score.

The predicted resistance/sensitivity to different therapeutics varied across morphotypes: MU resistance to gefitinib; DE sensitivity to azaticidine, dasatinib, and aplidin, and resistance to tamoxifen and gefitinib; PP resistance to cantharidin, SE resistance to aplidin, CT sensitivity to alkylating agents (*Figure 6*). The differences were observed even within tumor (*Figure 5*), with some of the supposedly sensitive tumors (whole tumor scoring) having regions of predicted resistance (*Figure 6—figure supplement 1*).

## Discussion

The analysis of the morphotypes from transcriptomics perspective is meant to bridge the histopathology and gene expression. The present exploratory study was motivated by our earlier observations linking the morphological aspects of CRC to the molecular subtypes (*Budinska et al., 2013*). The original observations semi-quantitatively scored the morphotypes as primary or secondary dominant in the whole tumor section and showed that subtype A (corresponding to CMS3) was enriched in PP and SE morphologies, subtype B (corresponding to CMS2) in CT morphology, subtype C (corresponding to CMS1) in MU and TB morphologies, and, finally, subtype D (CMS4) in DE/stromal reaction (*Budinska et al., 2013*). In contrast, here we focused on tumor regions rather than whole tumor, which also allowed the characterization of the intra-tumor heterogeneity.

The results show a whole landscape of changes at gene and pathway levels, with morphotypes residing on a continuum space of molecular descriptors. The analysis of hallmark pathways and

selected signatures combined with in silico deconvolution of cellular admixtures served two purposes. First, to confirm that the samples exhibit known properties (e.g. TB, SE, and PP have high tumor epithelial cell content, and DE and MU are enriched in fibroblasts; molecular EMT signature is high in MU and DE, but low in CT and SE, etc.), thus ensuring proper quality of the data. Second, it served to refine the characterization of the morphotypes and sketching their 'molecular portraits'. The morphotypes investigated had a fluid characterization from a transcriptomics perspective, with many pairwise similarities and some striking differences. Even from a strict histopathology perspective, it was difficult, if not impossible, to clearly distinguish the separation between adjacent morphological regions therefore a certain degree of contamination between morphotypes was to be expected. Nevertheless, the enrichment in specific cell types and states allowed the identification of characteristic molecular features.

MU and DE morphotypes (previously associated with CMS1 and CMS4, *Budinska et al., 2013*), as expected, exhibited high score of genes up-regulated in colon fibroblast TGF-β signaling pathway, genes associated with high tumor stromal content, CAFs and endothelial cells as well as pathways involved in immune system response. The detailed analysis of CAFs in fibroblast-rich regions (DE, MU, and ST) based on signatures derived from single cell sequencing studies (*Pelka et al., 2021*; *Kieffer et al., 2020*) revealed some finer differences: the supportive stroma (ST) region had the complete panel of fibroblast tested, while DE and MU most notably missed the 'normal CAFs'. The main difference between DE and MU appeared to be that former was enriched in CAFs associated with inflammatory response (IL-iCAF), all the other CAFs being present at similar levels. The other morphotypes were either significantly depleted in fibroblast signatures or their GSEA scores were not statistically significant. Deconvolution of immune cell fractions by quantiSeq showed enrichment of DE and MU in M1 macrophages. Given the involvement of CAFs in modeling the tumor microenvironment through ECM remodeling, angiogenesis promotion and immune system regulation (*Desbois and Wang, 2021*), our results support the idea of scoring separately the stromal component by either molecular or histopathology descriptors, in addition to tumor regions themselves. Even though DE and MU (and ST) also had the highest scores for the molecular EMT signature, our observations rather support the description of CMS4 as stromal/desmoplastic subtype than 'true' mesenchymal, in agreement with (*Loughrey et al., 2021*). Further, the poor prognostic associated with CMS4 could be explained by the stromal component: both (*Roseweir et al., 2020*) and (*Ten Hoorn et al., 2022*) agree that a high stromal invasion/desmoplastic reaction is prognostic of shorter time to relapse.

CT morphotype represents a classic adenocarcinoma and is one of the most common morphologies. In our previous study (*Budinska et al., 2013*), this morphotype was associated mainly with subtype B (vastly overlapping with CMS2). TB morphotype seems to be mostly representative of higher-grade tumors and was associated with CMS1. In contrast to NR, CT and TB showed significant enrichment of signatures of normal colon basal cells. From the molecular perspective, CT together with TB had both activated major pathways involved in proliferation processes. TB, in addition, resembled MU and DE morphotypes by sharing active TGF-β signaling, apoptosis, and active immune system response. SE and PP morphologies may be indicative of a different oncologic pathway – the 'serrated pathway' (*De Palma et al., 2019*). The two morphologies share common features like well to moderately differentiated, with low stromal content and crypt structure still preserved. From a molecular perspective, we found that both SE and PP were both distributed similarly across molecular subtypes (both CMS and iCMS) and had similar activation of hallmark pathways: EMT, *IL2/STAT5*, *IL6/STAT3*, *KRAS* signaling all being down-regulated, while *MYC* targets being up-regulated. Among the hallmark pathways, androgen response, heme metabolisms and *IL6/STAT3* (all silenced), appeared to be specific (and statistically significant) to SE and PP.

Given the relatively small sample size and similarities already observed between the morphotypes, it came as no surprise that the lists of differentially expressed genes, morphotype-specific, were generally short (for FDR ≤0.15). Nevertheless, literature search of the genes on top of these lists showed importance of these genes in CRC development, progression, EMT transition or response to therapy. For CT, the top gene was *PIP5K1B* which was related to PI3K/AKT signaling and seems to be involved in colorectal cancer development (*Zhang et al., 2019*). TB had the most differentially expressed genes (n=662) in comparison with all other morphotypes, with top genes including *FBXO5* – prognostic of shorter time-to-relapse in various cancers (*Liu et al., 2022*), *FLRT3* – a proapoptotic gene which, when overexpressed, inhibits EMT (*Yang et al., 2022*), *SETSIP* – gene coding chromatin-binding

protein capable of participating in fibroblast reprogramming and differentiation into epithelial cells (*Margariti et al., 2012*), *E2F7* – up-regulated by *p53* in response to DNA damage (*Carvajal et al., 2012*), *CXCL14* (downregulated) - depending on the cell of origin can have both tumor suppressive or supporting role (*Westrich et al., 2020*), *SEMA5A* (downregulated) gene – proposed as prognostic marker in CRC (*Demirkol et al., 2017*). Among top overexpressed genes specific to MU morphotype we found *FGF7* (fibroblast growth factor 7) whose disrupted signaling was associated with deregulation of cell differentiation (*Patel et al., 2019*), and *MUC2* (intestinal mucin) whose downregulation has been suggested a marker of adverse outcome (*Betge et al., 2016*). At the same time, *MUC2* was also among the DE-specific genes, but downregulated, consistent with the observation that desmoplastic reaction is a marker of shorter relapse-free survival (*Ueno et al., 2021*). Still in DE, we found as top overexpressed genes *PIEZO2* – a paralog of *PIEZO1* which is involved in colorectal cancer metastases (*Sun et al., 2020*), *SLIT3* – a member of the Slit/Robo pathway, a major regulator of several oncogenic pathways and potential therapeutic target (*Gara et al., 2015*), and *OLFML2B* – a potential biomarker for resistance to MEK inhibitors (*Hu et al., 2022*). SE morphotype had only one specific gene overexpressed at FDR ≤0.15, *CCDC175*. At the other end of the list, very interestingly, we found significantly downregulated gene for dihydropyrimidine dehydrogenase (*DPYD*) gene – the variants of which are predictive of 5-fluoruracil toxicity in adjuvant colon cancer treatment (*Lee et al., 2014*), *GLIPR2* which participates in positive regulation of ERK1/2 cascade and EMT transition (*Kang et al., 2012b*), or the *HOXA9A* gene, the overexpression of which was suggested to contribute to stem cell overpopulation responsible for development of CRC (*Osmond et al., 2022*) or the *GLI3* gene – that participates in sonic hedgehog (Shh)-Gli-mediated tumorigenesis and the loss of Gli3 signaling was shown to initiate cell growth inhibition in colon cancer cells, while sensitizing colon cancer cells to treatment with anticancer agents (5-FU and bevacizumab) (*Kang et al., 2012a*). The only specific gene marker of the PP morphotype was the downregulation of *MZP* – myelin protein zero.

We also found significant differences between pairs of morphotypes, especially in terms of molecular signatures/programs. These results reinforce the observations above and show that they are robust to the proportion of fibroblasts and/or epithelial cells present in the compared morphotypes.

In our collection, several cases were represented by several regions and an additional whole-tumor profile. Taking advantage of these matched samples, we investigated several molecular classifiers from an intra-tumor variability perspective as well. The CMS classification was less stable than iCMS, with whole tumor CMS class differing from at least one of the constituent morphological regions in about 60% of cases (11 out of 18, excluding cases in which CMS class was not predicted; see *Figure 5*). Additionally, we tested several prognostic and predictive expression-based classifiers/signatures. The goal was not to compare them in terms of their predictive capabilities (the experimental design did not allow for such an exercise), but rather to have a clear picture of the extent to which the various morphotypes 'distract' these predictors. We found that all the prognostic signatures varied with the morphological regions with some striking cases in which the morphotype scores exceeded the corresponding whole tumor scores by more than 50%. This observation suggests that, in some cases, the whole tumor-based predictions were too optimistic, the models failing to recognize higher risk cases. While these signatures were derived from whole-tumor expression profiles, their variability across tumor indicates the need for precise tumor sampling strategies.

Our exploratory study has, inherently, several limitations. The selection of cases may not represent the proportions of various morphotypes found in general population of CRC patients. Our selection tried to cover as many scenarios as feasible with a limited number of samples. Also, the tumor heterogeneity in terms of morphotypes cannot be estimated from these data since a single tissue block per tumor was considered. The reduced sample size in some of the paired comparisons within same tumor calls for further external validation. However, our results pave the way to future studies addressing these questions and others related to optimizing the tumor sampling strategy, for example.

We have analyzed the gene expression profiles of six morphotypes (and two peritumoral regions), building a comprehensive molecular picture of their salient features. The observed heterogeneity, especially intra-tumoral (*Figure 5*), calls for a finer resolution of the tumor sampling in profiling studies. Until spatial transcriptomics becomes integrated in routine clinical practice, using the morphotypes for anchoring the expression profiles is a feasible approach. Our study already provides indications of the molecular programs one would expect to find de-/activated in these regions, thus helping in designing future experiments. The implications for molecular classifiers are clear: it is necessary to

account for tumor morphology when designing new biomarkers. Given the sensitivity of many gene-based classifiers to the tumor and stroma proportions in the samples, there is a need to adjust these classifiers to control for their relative proportions. This can be achieved by different means, and we presented an approach based on morphotypes.

From a molecular pathology practice perspective, the molecular descriptors found to vary across morphotypes may help in patient stratification and provide hints for further, more targeted investigations. Several questions call for further investigation: (i) how much of a tumor needs to be embedded to achieve a precise molecular diagnostic? and (ii) what precise tumor region(s) are needed for a molecular diagnostic? The morphotypes selected here may need further refinement and achieving consensus among pathologists regarding their exact definition, a point that could potentially be addressed by automatic image analysis approaches.

### Ideas and speculation

Our analyses indicate that both prognostic and response to therapy signatures may predict more severe cases (shorter relapse free survival or resistance to therapy) when applied to subregions than to the whole tumor. This might be one of the reasons the said signatures may fail their real-world validation. Therefore, morphologically heterogeneous tumors need several sampling locations to provide a more sensible result. Sensitivity and cost analyses need to be performed to estimate the benefits of multi-regional sampling.

Further, the fact that we were able to identify specific molecular programs associated with the morphotypes calls for investigating the inverse problem as well, that is whether sufficiently discriminatory features could be extracted for estimating the proportions of the morphotypes from whole tumor profiles.

## Materials and methods
### Samples

This retrospective cross-sectional study used tumor samples from patients with CRC who were examined at Masaryk Memorial Cancer Institute, Brno, Czech Republic in years 2002–2015. The study was reviewed and approved by the Committee for Ethics of Masaryk Memorial Cancer Institute, Brno, Czech Republic (number 2018/861/MOU). All patients gave written informed consent for the use of their biological samples for research purposes. Fundamental ethical principles and rights promoted by the European Union EU (2000/C364/01) were followed. All patients' data were processed according to the Declaration of Helsinki (last revision 2013). Inclusion criteria for this study were: age >18 years, clinical and histopathologically confirmed diagnosis of primary CRC. Standard clinical and histopathological variables (TNM, grade etc.) were retrieved for all patients. Failure of laboratory analyses (problematic sample preparation, low quality and/or quantity of isolated RNA, low quality of expression data) was a reason for excluding these samples from the study.

### Sample preparation

A total of 111 colon cancers (unique patients) were identified in the tumor archive of the Masaryk Memorial Cancer Institute and were assessed by two expert pathologists. Morphological regions of interest, representing complex tubular (CT), desmoplastic (DE), mucinous (MU), papillary (PP), serrated (SE) and solid/trabecular (TB) morphologies, respectively (see *Figure 1*), were digitally marked in scanned whole slide images (at 20 x magnification) and macrodissected for RNA extraction. Additionally, from several slides, tumor-adjacent normal (NR) and tumor-associated stroma (ST). Tumor samples with limited contamination of additional morphologies (<20%) were called 'core samples' and used morphotype molecular characterization. The labelling of the regions was repeated after 1 year to ensure a stable assignment. For n=28 cases, whole-tumor regions were macrodissected from the histology section immediately adjacent to the section used for morphological regions. Standard clinical and histopathological variables were retrieved for most of the patients.

### Gene expression profiling

The RNA extraction was performed from formalin-fixed paraffin-embedded histopathological slides using AllPrep DNA/RNA Kits (Qiagen, Hilden, Germany) according to their specific manufacturer's

instructions. A few modifications were made to the protocol: FFPE slides (2x3 μm) were bathed in a solution to remove paraffin (3 x in xylene for 5 min and 3 x in ethanol for 5 min). Tumor tissue was spotted with 8 ul PKD puffer and collected from slides using a scalpel. Purification was done for total RNA, including small RNAs. For elution, 20 ul RNA free water (1 min. incubation) was used and then repeated with eluate. The extracted RNA served as input for a GeneChip WT Pico Reagent Kit (Thermo Fisher Scientific, Waltham, MA, USA) for analysis of the transcriptome on whole-transcriptome arrays. We selected the input amount from the recommended range according to the manufacturer's instructions. Total RNA from HeLa cells provided in the kit was used as a positive control together with a high-quality low-concentration RNA isolated from a serum as a low input control. Clariom D Array for human samples (Thermo Fisher Scientific, Waltham, MA, USA) was used for target hybridization to capture both coding and multiple forms of non-coding RNA. Finally, the arrays were scanned using Affymetrix GeneChip Scanner 3000 7 G (Thermo Fisher Scientific, Waltham, MA, USA). The sample preparation and analysis were performed according to the manufacturer's instructions. The protocol included several control points in which the workflow was monitored. All the samples complied with the quality control requirements and none of the samples were excluded from the analysis.

The data generated in this study are publicly available in ArrayExpress under accession number E-MTAB-12599 (https://www.ebi.ac.uk/biostudies/arrayexpress/studies/E-MTAB-12599).

### Bioinformatics analyses

All resulting CEL files were processed using Bioconductor (RRID:SCR_006442) (*Huber et al., 2015*) (v.3.15) packages oligo (*Carvalho and Irizarry, 2010*) (v.1.60), affycoretools (v1.68) and, for Clariom D chip annotation, pd.clariom.d.human (v.3.14). For the quality control we used AffyPLM (v.147) and imposed a maximal median Normalized Unscaled Standard Errors (NUSE) of 1.12. In all, n=202 passed all the quality control steps and were normalized together using RMA (oligo) with core-probeset summarization. Further, the array data was summarized at gene level by selecting the most variable probeset per unique EntrezID and entries corresponding to missing HUGO symbols, speculative transcripts, and short non-coding RNA were discarded resulting in a reduced list of 27,302 unique genes. Batch effects were removed using ComBat (*Johnson et al., 2007*) from package sva (v.3.44.0).

For the identification of differentially expressed genes we used linear models (limma package v.3.52.2) with a cut-off for false discovery rate FDR = 0.15. The pathways were scored in terms of enrichment in specific signatures using gene set enrichment analysis (GSEA) (*Subramanian et al., 2005*) as implemented in fgsea package (v.1.22.0). For scoring the signatures in individual samples, we used gene score variation analysis (GSVA) (*Hänzelmann et al., 2013*) implemented in GSVA package (v.1.44.1). MSigDB (RRID:SCR_016863) (all collections: H, C1-8; v.7.4.1) (*Liberzon et al., 2015*) was used as the main source for gene sets and pathways. Additional cell type-specific gene sets, some derived from whole tumor others from single-cell sequencing studies, representing (i) cancer associated fibroblasts (CAFs) (*Isella et al., 2015*; *Pelka et al., 2021*; *Khaliq et al., 2022*; *Kieffer et al., 2020*) (ii) epithelial cells (*Kosinski et al., 2007*; *Merlos-Suárez et al., 2011*; *Pelka et al., 2021*), and (iii) immune cells (*Isella et al., 2015*; *Pelka et al., 2021*) were used (see *Supplementary file 3* for full list). The consensus molecular subtypes were predicted using CMSCaller (*Eide et al., 2017*) (v.2.0.1) and the intrinsic epithelial subtypes (*Joanito et al., 2022*) using the signatures therein (P. Tsantoulis, personal communication, July 2022). The cellular mixture of various tumoral regions was explored computationally using quanTIseq (*Finotello et al., 2019*) (for immune cells) and ESTIMATE (*Yoshihara et al., 2013*) (for tumor purity/epithelial cells). The core samples were used for deriving the lists of differentially expressed genes, for gene set enrichment analyses and for in silico deconvolutions of cell populations. The analyses treating the samples independently were applied to all samples, including non-core.

Ten different survival/prognostic genomic signatures (full list in *Supplementary file 13*) were computed per-sample as (weighted, when weights were provided) means of signature genes, and 29 sensitivity/resistance signatures selected from MSigDB/C2 were scored by GSVA.

All data analyses were performed in R 4.2 (*R Development Core Team, 2022*).

### Acknowledgements

We thank Dr. Petros Tsantoulis, Hôpitaux Universitaires de Genève, Switzerland, for providing us with a model for classification of intrinsic epithelial subtypes based on gene expression data. The authors

acknowledge funding from the Czech Science Foundation (GACR) grant no. GA19-08646S. Also, the support of the Research Infrastructure RECETOX RI (No LM2018121) and Cetocoen Plus project (CZ.0 2.1.01/0.0/0.0/15_003/0000469) financed by the Ministry of Education, Youth and Sports of the Czech Republic (MEYS), the Teaming project (CETOCOEN Excellence 857560; CZ.02.1.01/0.0/0.0/17_043/0 009632), and the project National Institute for Cancer Research (Programme EXCELES, ID Project No. LX22NPO5102) - Funded by the European Union - Next Generation EU is acknowledged. This work was supported from the European Union's Horizon 2020 research and innovation program under grant agreement No 857560. This publication reflects only the author's view, and the European Commission is not responsible for any use that may be made of the information it contains.

## Additional information

### Funding

| Funder | Grant reference number | Author |
|---|---|---|
| Grantova Agentura Ceske Republiky | GA19-08646S | Eva Budinská<br>Martina Hrivňáková<br>Vlad Popovici |

The funders had no role in study design, data collection and interpretation, or the decision to submit the work for publication.

### Author contributions

Eva Budinská, Conceptualization, Data curation, Formal analysis, Validation, Visualization, Methodology, Writing – original draft, Project administration, Writing – review and editing; Martina Hrivňáková, Data curation, Methodology, Writing – review and editing; Tina Catela Ivkovic, Lenka Zdražilová Dubská, Data curation, Validation, Writing – original draft, Writing – review and editing; Marie Madrzyk, Data curation, Software, Writing – original draft; Rudolf Nenutil, Resources, Methodology, Writing – original draft; Beatrix Bencsiková, Data curation, Formal analysis, Writing – original draft, Writing – review and editing; Dagmar Al Tukmachi, Data curation, Methodology, Writing – original draft, Writing – review and editing; Michaela Ručková, Software, Validation, Writing – original draft; Ondřej Slabý, Supervision, Investigation, Methodology, Writing – original draft; Josef Feit, Supervision, Validation, Investigation, Writing – original draft; Mihnea-Paul Dragomir, Formal analysis, Investigation, Writing – original draft, Writing – review and editing; Petra Borilova Linhartova, Investigation, Writing – original draft, Writing – review and editing; Sabine Tejpar, Supervision, Investigation, Writing – original draft; Vlad Popovici, Conceptualization, Supervision, Funding acquisition, Investigation, Methodology, Writing – original draft, Project administration, Writing – review and editing

### Author ORCIDs

Vlad Popovici http://orcid.org/0000-0002-1311-9188

### Ethics

Human subjects: This retrospective cross-sectional study used tumor samples from patients with CRC who were examined at Masaryk Memorial Cancer Institute, Brno, Czech Republic in years 2002-2015. The study was reviewed and approved by the Committee for Ethics of Masaryk Memorial Cancer Institute, Brno, Czech Republic (number 2018/861/MOU). All patients gave written informed consent for the use of their biological samples for research purposes. Fundamental ethical principles and rights promoted by the European Union EU (2000/C364/01) were followed. All patients' data were processed according to the Declaration of Helsinki (last revision 2013). Inclusion criteria for this study were: age > 18 years, clinical and histopathologically confirmed diagnosis of primary CRC. Standard clinical and histopathological variables (TNM, grade etc.) were retrieved for all patients. Failure of laboratory analyses (problematic sample preparation, low quality and/or quantity of isolated RNA, low quality of expression data) was a reason for excluding these samples from the study.

Joint Public Review: https://doi.org/10.7554/eLife.86655.3.sa1
Author Response https://doi.org/10.7554/eLife.86655.3.sa2

## Additional files

### Supplementary files
• Supplementary file 1. Main clinical parameters of the study cohort.
• Supplementary file 2. Distribution of main clinical parameters per morphotype (and tumor-adjacent normal and supportive stroma).
• Supplementary file 3. Table of gene expression signatures.
• Supplementary file 4. Table of GSEA scores (NES) for "other" signatures (see also *Supplementary file 3* for signatures).
• Supplementary file 5. List of differentially expressed genes (limma tables) per morphotype in contrast with pooled profile.
• Supplementary file 6. GSEA results for genes in *Supplementary file 5*, for whole MSigDB collection.
• Supplementary file 7. List of differentially expressed genes (limma tables) per morphotype in contrast with all other five morphotypes.
• Supplementary file 8. GSEA results for genes in *Supplementary file 7*, for whole MSigDB collection.
• Supplementary file 9. List of differentially expressed genes (limma tables) per matched pairs of morphotypes.
• Supplementary file 10. GSEA results for genes in *Supplementary file 9*, for whole MSigDB collection.
• Supplementary file 11. List of differentially expressed genes (limma tables) for pairs of morphotypes.
• Supplementary file 12. GSEA results for genes in *Supplementary file 11*, for whole MSigDB collection.
• Supplementary file 13. List of prognostic signatures tested.
• MDAR checklist

### Data availability
Data generated through this study is publicly available from ArrayExpress under accession number E-MTAB-12599. All analyses results are available as supplementary files.

The following dataset was generated:

| Author(s) | Year | Dataset title | Dataset URL | Database and Identifier |
|---|---|---|---|---|
| Popovici V, Budinska E | 2023 | Molecular portraits of colorectal tumor morphological regions | https://www.ebi.ac.uk/biostudies/arrayexpress/studies/E-MTAB-12599 | ArrayExpress, E-MTAB-12599 |

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
