## [Editor Report · eLife assessment]

This study presents a **valuable** finding on the putative molecular patterns underlying characteristic morphological regions observed in colorectal cancer (CRC). The authors provide a morphological framework through which clinicians might improve the performance of molecular signatures and consequently predict the clinical response of patients with better accuracy. The evidence supporting the claims of the authors is **solid**. The work will be of interest to clinicians and cancer biologists working in the field of CRC.

---

## [Referee Report · Joint Public Review]

The manuscript by Budinska et al investigated that morphological heterogeneity may have an impact on gene-expression profiles and conventional molecular signatures applied to bulk CRC tissues. The authors conducted whole transcriptome microarrray profiling data from macro-dissected morphotype-specific tumor regions, bulk tumor and surrounding normal and stromal tissues to support their claims. The paper is interesting as it provides a putative morphological approach through which clinicians might improve the performance of molecular signatures and consequently predict the clinical response of patients with better accuracy. In the updated version of the manuscript, the authors have improved the manuscript and addressed several unsolved concerns such as patient selection and tumor area selection to justify their claims. The findings of the manuscript may have potential to be translated into the clinic of CRC.

---

## [Author Response]

The following is the authors’ response to the original reviews.

We are grateful to the reviewers for their insightful comments, suggestions, and criticism. In the updated version of the manuscript, all these will be properly reflected. Here we briefly address the main points raised:

**Reviewer #1:**
(1.1) Patient selection and tumor area selection are crucial for this study but not very carefully defined. Why are some core and others not? Figure referral is an issue here (sup figure 6 where all core and non-core samples are supposed to be according to the legend of Fig 4 is likely sup fig 7 but this is then a complete copy paste of Figure 4). In the methods it is stated that the core samples are based on limited contamination of additional morphotypes (<20%) but Fig 4 suggests that all tumours listed have multiple morphotypes.

The tissue samples were obtained from a hospital cohort of patients with stage II-IV colorectal cancer (at diagnostic time), with no particular selection criteria imposed, as this was an exploratory study.

Tumor regions were marked for macro-dissection by an experienced pathologist following the standard practice for whole-tumor transcriptomics studies. The subregions (morphological regions) were marked by the same experienced pathologist for macro-dissection (in an adjacent section) and reassessed later with respect to their “morphological purity”. It is impossible to macro-dissect regions containing a single morphological pattern. Hence, those regions which contained significant amount (>=20%) of other morphologies were considered “non-core”, while the rest were called “core” regions. This distinction applies to morphological regions solely and not to whole-tumor samples.Indeed, the reference in caption to Figure 4, should refer to Supp. Fig. 7 (and has been updated).

(1.2) CMS subtype should be performed with single sample predictor rather than CMScaller.

We agree that a single-sample predictor for CMS is needed, however CMScaller is the de facto classifier for CMS (>130 citations) so we used it to illustrate the practical implications.

(1.3) A couple of surprising observations need specification. MUC2 is a strong CMS3 reporter gene yet Mucinous tumours appear to end up in CMS4 rather than 3. Can the authors show that indeed stroma cells are very evident in these samples?

We do not have a direct estimation of the amount of stromal cells, but the high scores of the various fibroblast-related signatures in mucinous regions (Fig2 B, D) indicate that, indeed, there is an enrichment in stroma. In the follow-up study we plan to perform specific staining as well as spatial transcriptomics of these regions to further investigate our findings.

(1.4) The SE PP and CT are assigned to CMS2, but in Figure 4 this appears a lot more variable than the authors would make the reader believe. The full data are not completely clear (see point 1).

In the paper, we transparently state that PP, SE, and CT were assigned to CMS2 in 62.5%, 41.7% and 41.9% of cases, respectively. These proportions referred to all samples for which CMSCaller made a prediction. In Fig.4, we also show the proportion of cases in which CMSCaller did not predict any subtype.

(1.5) The tumor response rates are rather weird as this is likely dependent on the complete tumour and not so much the subareas. It is not very well described what we see in this analysis.

We did not compute any response rates but simple prognostic scores as (weighted, if weights were provided) means of genes in the specific signatures (see Methods). The question addressed was whether these scores were comparable between whole tumor and corresponding tumor regions (within same tumor). Given the observed (relative) variability, the more important follow-up question - which we cannot answer with our limited survival data – is whether a higher score in a region in comparison with whole-tumor is indeed indicative of a higher risk of relapse.

(1.6) Serrated adenomas have previously been aligned with CMS4. Is this different from serrated areas in cancers?

We do not have data from adenomas to compare with the serrated carcinoma regions. But a comparison of (regions of) both traditional serrated and sessile serrated adenomas to serrated carcinoma would be interesting.

(1.7) The fact that iCMS2 and iCMS3 align rather well with the current analysis of the distinct regions suggests that the analysis that was reported last year is the proper way to view tumor intrinsic signatures. The authors now propose a rather similar outcome to this issue which does take away a lot of the novelty of the findings of this study.

In the manuscript it is clearly stated that our goal was to describe the molecular characteristics associated with several morphological patterns. It was not to propose another stratification paradigm for colorectal cancer. As such, our analyses were not limited to molecular subtypes and the respective observations were but a small part of our findings. Indeed, the intrinsic subtypes (iCMS 2/3) were stable and robust, as they were based on the genes expressed in epithelial cells, and they might well prove to be of clinical importance too. However, they do not cover all aspects (e.g. fibroblasts subtypes) and, as stated in Joanito et al. Nat Gen 54, pages 963–975 (2022), “iCMS, MSI status and CMS jointly inform the molecular classification of CRC”. Last, in our opinion, the molecular classification of CRC, while a useful point of view in tumour classification, is not covering all the necessary perspectives on tumour heterogeneity.

**Reviewer #2:**
(2.1) Overall, the manuscript provides an interesting histological/morphological framework through which we can consider heterogeneity in colorectal carcinoma and an approach by which we might improve the performance of gene expression-based classifiers in predicting clinical behaviour and/or responses to therapy. Exploration of CRC morphotypes and their differences was quite interesting. However, more work is needed to support the claims made by the authors. While I appreciate that the authors themselves identify limitations of their study within the manuscript, I believe awareness of these limitations is not reflected in some of the claims made in the abstract and at points in the main text when discussing the use of expression-based classifiers.

The manuscript was improved to clarify several aspects that Reviewer 2 rightly pointed out:

1. We clarify that for a patient (tumor) there might be one or several corresponding transcriptomics profiles (see Methods).

2. The resulting “molecular portraits” were not derived with the goal to deconvolve the bulk tumor expression profiles and to estimate the proportions of morphotypes. Whether this is possible at all, is an open question and we mention this aspect in “Ideas and Speculation” section.

3. We improved figures captions to be more descriptive.

4. We included the reference for “Isela signature” at its first appearance.